# MobileCLIP2: Improving Multi-Modal Reinforced Training

**Fartash Faghri**[*]                                                     *fartash@apple.com*
*Apple*

**Pavan Kumar Anasosalu Vasu**[*]                           *panasosaluvasu@apple.com*
*Apple*

**Cem Koc**                                                              *cem_koc@apple.com*
*Apple*

**Vaishaal Shankar**
*Work done while at Apple*

**Alexander Toshev**                                                    *toshev@apple.com*
*Apple*

**Oncel Tuzel**                                                          *otuzel@apple.com*
*Apple*

**Hadi Pouransari**                                                 *mpouransari@apple.com*
*Apple*

**Reviewed on OpenReview:** *https://openreview.net/forum?id=WeF9zolng8*

## Abstract

Foundation image-text models such as CLIP with zero-shot capabilities enable a wide array of applications. MobileCLIP is a recent family of image-text models at 3–15ms latency and 50–150M parameters with state-of-the-art zero-shot accuracy. The main ingredients in MobileCLIP were its low-latency and light architectures and a novel multi-modal reinforced training that made knowledge distillation from multiple caption-generators and CLIP teachers efficient, scalable, and reproducible. In this paper, we improve the multi-modal reinforced training of MobileCLIP through: 1) better CLIP teacher ensembles trained on the DFN dataset, 2) improved captioner teachers trained on the DFN dataset and fine-tuned on a diverse selection of high-quality image-caption datasets. We discover new insights through ablations such as the importance of temperature tuning in contrastive knowledge distillation, the effectiveness of caption-generator fine-tuning for caption diversity, and the additive improvement from combining synthetic captions generated by multiple models. We train a new family of models called MobileCLIP2 and achieve state-of-the-art ImageNet-1k zero-shot accuracies at low latencies. In particular, we observe 2.2% improvement in ImageNet-1k accuracy for MobileCLIP2-B compared with MobileCLIP-B architecture. Notably, MobileCLIP2-S4 matches the zero-shot accuracy of SigLIP-SO400M/14 on ImageNet-1k while being 2× smaller and improves on DFN ViT-L/14 at 2.5× lower latency. We release our pretrained models [1] and the data generation code [2]. The data generation code makes it easy to create new reinforced datasets with arbitrary teachers using distributed scalable processing.

---

[*]Equal contribution.
[1]https://github.com/apple/ml-mobileclip
[2]https://github.com/apple/ml-mobileclip-dr

# 1 Introduction

CLIP (Radford et al., 2021) is an image-text model that maps images and text inputs to a shared embedding space, where a text describing an image, also called caption, is mapped close to an image matching its description but far from dissimilar images. Building on a vast literature (Frome et al., 2013; Socher et al., 2014; Karpathy & Fei-Fei, 2015; Kiros et al., 2014; Faghri et al., 2018), CLIP substantially increased the scale of training data and models. Consequentially, along with improved image-text retrieval performance, new zero-shot classification capabilities emerged with non-trivial accuracy on classification tasks without any explicit supervised training with classification labels through linear probing. The image-encoder can be further specialized to a new task by either linear probing (fixed encoder), or full fine-tuning to achieve state-of-the-art performance on a diverse set of tasks (Wortsman et al., 2022). CLIP is one of the first to be called a foundation model given the diversity of its capabilities and applications (Bommasani et al., 2021).

The success of CLIP resulted in an increase in the sizes of models and datasets, leading to a gradual increase in performance (Fang et al., 2024b; Zhai et al., 2023; Gadre et al., 2023; Fang et al., 2024a). Recently, this trend has been reversed to models with small size and low latency for applications on mobile devices. Notably, TinyCLIP (Wu et al., 2023) and MobileCLIP (Vasu et al., 2024c) proposed models with as few as 50M total parameters (sum of image and text encoder parameters). For example, MobileCLIP-S0, with total latency of 3ms (sum of image and text encoder latencies), achieves similar average performance to the original OpenAI ViT-B/16 CLIP while being 3x smaller and 5x faster. It also demonstrates improved performance compared to prior state-of-the-art larger models, such as SigLIP (Zhai et al., 2023).

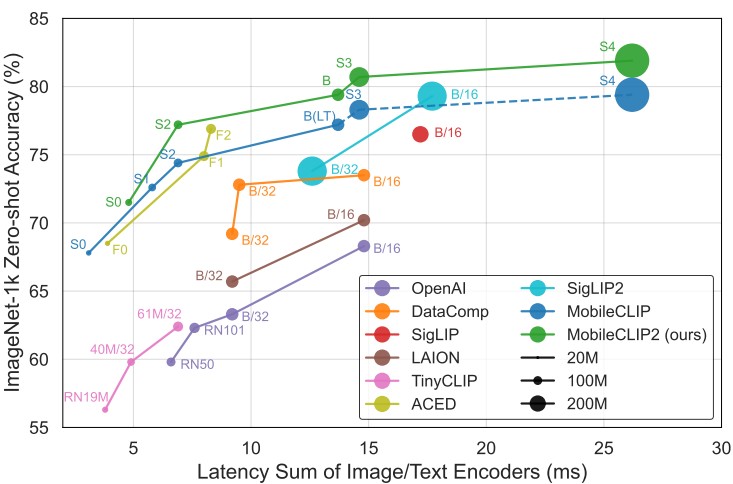

Figure 1: **MobileCLIP2 models trained on DFNDR-2B achieve state-of-the-art accuracy at low latencies.** MobileCLIP2-S4 matches the accuracy of SigLIP-SO400M/14 with 2× fewer parameters and surpasses DFN ViT-L/14 at 2.5× lower latency measured on iPhone12 Pro Max. MobileCLIP-S3/S4 are our new architectures trained on MobileCLIP's training dataset, DataCompDR-1B (dashed lines).

In this paper, we present ablations of multi-modal reinforced training and present an improved training recipe. We train a new family of models, MobileCLIP2, that establishes new state-of-the-art ImageNet-1k accuracy at a range of latencies matching the performance of larger SigLIP (Zhai et al., 2023) and DFN (Fang et al., 2024a) models while up to 4× smaller (our MobileCLIP2-S2 compared with SigLIP2-B/32) and up to 2.5× faster (our MobileCLIP2-S4 compared with DFN ViT-L/14). Moreover, we release efficient distributed code for generating reinforced datasets using arbitrary teacher models.

# 2 Improved Training

MobileCLIP introduced a family of low-latency image-text models consisting of S0, S1, S2, B, and B-LT variants with aggregate image-text latencies spanning 3.8-13.7ms. These low latencies were achieved with specialized architectures based on FastViT (Vasu et al., 2023b) and an improved training method called multi-modal reinforced training. We seek to further explore and improve each step of multi-modal reinforced training. We additionally consider a more diverse family of architectures that cover a wider range of latencies.

Reinforced training is a method for achieving better performance from a base dataset through improvements from additional sources such as pretrained models (Faghri et al., 2023). Multi-modal reinforced training introduced in Vasu et al. (2024c) adds information to an image-text dataset from pretrained image-text models

Table 1: **Summary of MobileCLIP2 training improvements.** CoCa models are pretrained on a large dataset for 13B seen samples then fine-tuned for 12M seen samples (denoted by →). The architecture for all CLIP teachers in this table is ViT-L/14. We report mean and standard deviations of 5 runs when available.

| Name | Dataset | CLIP Teacher Datasets | CoCa Dataset | IN-val | Flickr30k | Avg. 38 |
|---|---|---|---|---|---|---|
| MobileCLIP (Vasu et al., 2024c) | DataComp-1B12M | OpenAI + DataComp-XL | LAION-2B → MSCOCO-123k | 61.6 | 72.8 | 53.5 |
| Table 2 | DFN-5B12M | OpenAI + DataComp-XL | LAION-2B → MSCOCO-123k | $63.1_{0.2}$ | $73.3_{0.6}$ | $54.1_{0.4}$ |
| Table 4 | DFN-5B12M | DFN-2B + DFN-2B-s39B | LAION-2B → MSCOCO-123k | $65.4_{0.4}$ | $75.8_{0.3}$ | $56.2_{0.6}$ |
| MobileCLIP2 (Tab. 6) | DFN-5B12M | DFN-2B + DFN-2B-s39B | DFN-2B → MSCOCO-38k | $65.9_{0.3}$ | $75.4_{0.2}$ | $56.5_{0.3}$ |
| Table 6 | DFN-5B12M | DFN-2B + DFN-2B-s39B | DFN-2B → Syn.×10 | $66.0_{0.1}$ | $75.1_{0.6}$ | $56.5_{0.3}$ |

as well as a pretrained synthetic caption generator. Specifically, they add the following additional information to DataComp-1B dataset: 1) image embeddings from two CLIP teachers on 10 random augmentations of each image 2) text embeddings from two CLIP teachers on the original text as well as 5 synthetic captions generated from a CoCa caption generator. Given a reinforced dataset, they modify the training loss to include a knowledge distillation loss given the embeddings from teachers on each sample (Hinton et al., 2015). To ensure consistency between the teacher and student, the same image augmentation is reproduced via stored augmentation parameters (Beyer et al., 2022; Faghri et al., 2023). They perform ablations to find the set of CLIP teachers, caption generator, and image augmentations that provide the largest performance gain on ImageNet as well as the average accuracy on 38 evaluations from DataComp (Gadre et al., 2023).

We follow a similar multi-modal reinforced training to MobileCLIP while improving all aspects and call the resulting model family MobileCLIP2. Table 1 summarizes the gains from each major improvement. In short, we use better training data, better CLIP teacher models, and better and diverse synthetic caption generators compared to MobileCLIP. In all ablations, we train MobileCLIP-B for 30k iterations (∼20 epochs) on datasets with 12.8M images. We provide a summary of datasets in this paper in Tab. 15.

Figure 2 demonstrates the efficiency gains compared with DFN (Fang et al., 2024a), DataComp (Gadre et al., 2023) and DataCompDR (Vasu et al., 2024c) datasets during training. Training on DFNDR-2B12M for 30M seen samples is 5x more efficient than training on DataComp-1B12M, i.e., we reach the ImageNet-1k zero-shot accuracy of training on DataComp-1B12M for 30M after seeing only 6M samples of DFNDR-2B12M. Similarly, training on DataCompDR-12M is 3.3x more efficient compared to DFN-2B12M and 1.3x more efficient compared with DataCompDR-12M. We also observe 1.6x speedup when training on DFNDR-2B compared with training on DataCompDR-1B for 13B seen samples. Similar to DataCompDR, training on DFNDR datasets do not have any wall-clock time overhead, i.e., each training step of training on DataComp, DFN, DataCompDR, and DFNDR takes the same amount of time. That means any efficiency gains in terms of the number of samples and training iterations directly translate to wall-clock time efficiency gains.

## 2.1 Multi-Modal Reinforced Training

Dataset Reinforcement (DR) (Faghri et al., 2023) is a method for improving a dataset to achieve higher accuracy with minimal changes to the training code and minimal computational overhead. DR was first introduced for training image classifiers where Faghri et al. (2023) improved the ImageNet dataset by storing classification probabilities efficiently from a strong ensemble of classifiers. Given stored probabilities, the training was essentially Knowledge Distillation (Hinton et al., 2015) with no overhead for computing the teacher predictions. The cost efficiency makes it feasible to train longer for larger gains as observed in Beyer et al. (2022). Vasu et al. (2024c) adopted DR for training image-text CLIP models by storing knowledge from a strong ensemble of CLIP models and additionally synthetic captions from an image caption generator. They demonstrated up to 1000× improved learning efficiency compared with non-reinforced CLIP training.

Given a batch of $b$ image-text pairs, we denote the embeddings of a target student model by $\Phi_{\text{img}}, \Phi_{\text{txt}} \in \mathcal{R}^{b \times d}$, where $d$ is the dimensionality of the shared embedding space. We utilize two types of teachers, an image-text teacher ensemble that maps images and texts to a shared space similar to CLIP (Radford et al., 2021) and CoCa-based captioners that generate a caption given an image using an encoder-decoder architecture (Yu et al., 2022). Let $\Psi_{\text{img}}^{(k)}, \Psi_{\text{txt}}^{(k)} \in \mathcal{R}^{b \times d_k}$ denote the image-text embeddings from the $k$-th CLIP-based teacher

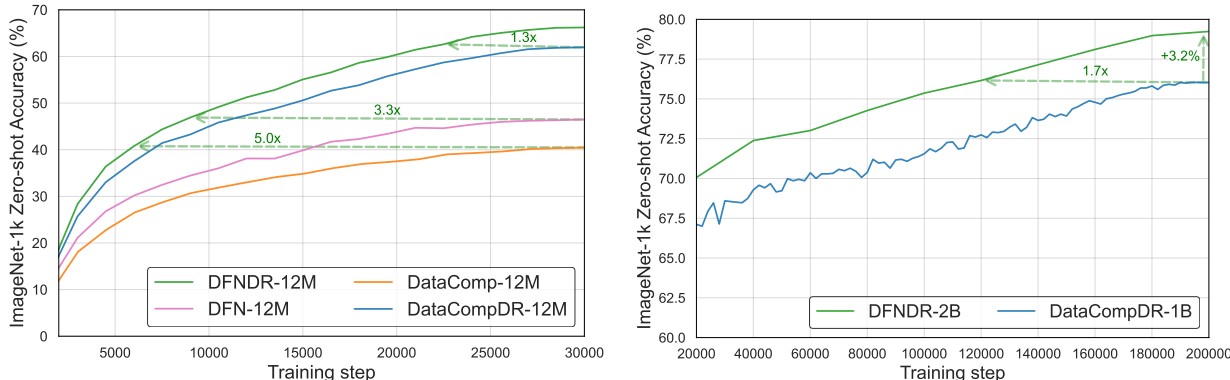

Figure 2: Left: **Training on DFNDR-12M is up to 5x more efficient** compared with DataComp-1B12M, 3.3x compared with DFN-12M, and 1.3x compared to DataCompDR-12M. All models are trained for 30k iterations and global batch size 8192 (246M seen samples). DFN-12M consists of 12M uniformly sampled image-text pairs from DFN-2B and DFNDR-12M consists of additional reinforcements per sample in DFN-12M. Right: **Training on DFNDR-2B is up to 1.7x more efficient** compared with DataCompDR-1B. Models are trained for 200k iterations and gloabl batch size 65536 (13B seen samples).

where $d_k$ is the dimensionality of the shared space. The distillation loss is defined as

$$\mathcal{L}_{\text{Distill}} = \frac{1}{2bK} \sum_{k=1}^{K} \underbrace{\text{KL}(\mathcal{S}_{\tau_k}(\Psi_{\text{img}}^{(k)}, \Psi_{\text{txt}}^{(k)}) \| \mathcal{S}_{\widehat{\tau}}(\Phi_{\text{img}}, \Phi_{\text{txt}}))}_{\text{Image to Text}} + \underbrace{\text{KL}(\mathcal{S}_{\tau_k}(\Psi_{\text{txt}}^{(k)}, \Psi_{\text{img}}^{(k)}) \| \mathcal{S}_{\widehat{\tau}}(\Phi_{\text{txt}}, \Phi_{\text{img}}))}_{\text{Text to Image}}, \qquad (1)$$

where KL denotes Kullback-Leibler divergence, and $\mathcal{S}_\tau(\boldsymbol{U}, \boldsymbol{V})$ is the row-wise Softmax operation applied to $\boldsymbol{U}\boldsymbol{V}^\top/\tau$ with temperature $\tau$. The total loss is $\mathcal{L}_{\text{Total}} = (1-\lambda)\mathcal{L}_{\text{CLIP}} + \lambda\mathcal{L}_{\text{Distill}}$, that is the sum of a standard CLIP loss and the distillation loss with coefficients $1-\lambda$ and $\lambda$, respectively.

## 2.2  Better Base Dataset: DFN

Multi-modal reinforced training starts from a base dataset containing real image-text pairs commonly collected from the web. DataComp (Gadre et al., 2023) demonstrated that the quality of large-scale image-text datasets can be significantly improved through filtering based on scores such as compatibility of image and text. Their proposed BestPool filtering applied on a pool of 12B samples resulted in the DataComp-1B dataset that was used as the base dataset in MobileCLIP. DataComp also released the original 12B samples as a benchmark for dataset curation and filtering methods. DFN (Fang et al., 2024a) proposed to filter data using a filtering network trained on high-quality data. Applying their model on DataComp-12B pool resulted the DFN-2B dataset. They additionally collected a larger set of images from the web disjoint from DataComp-12B and after filtering resulted in another 3B samples and collectively created the DFN-5B dataset.

We study the impact of replacing the base dataset in MobileCLIP with DFN-5B. We ablate using 12M uniformly sampled subset of DataComp-1B referred to as DataComp-1B12M that was introduced in (Vasu et al., 2024c) for rapid experimentation. We similarly sample a 12M subset from DFN-5B referred to as DFN-5B12M. Table 2 compares the performance of training with and without distillation/synthetic captions. We observe that DFN-5B12M results in up to 1.4% gain together with distillation and synthetic captions. Although this gain is smaller compared to the up to 6% gain without distillation/synthetic captions, it is still more than standard deviation.

## 2.3  DFN CLIP Teachers

One source of reinforcement in multi-modal reinforced training is the embeddings from CLIP teachers that are used as targets in CLIP distillation. (Vasu et al., 2024c) performed a comprehensive study of existing strong CLIP teachers at the time of publication and found the ensemble of `ViT-L-14-openai`

Table 2: **Training on DFN is better than DataComp with and without distillation/synthetic captions.** CLIP teachers and synthetic caption generators are the same as MobileCLIP (OpenAI+DataComp-XL CLIP-ViT-L/14, and CoCa-ViT-L/14). For distillation, the coefficient $\lambda$ is set to 1.0 (no CLIP loss) and use strong image augmentations.

| Dataset | Distill. | Syn. Caps. | IN-val | Flickr30k | Avg. 38 |
|---|---|---|---|---|---|
| DataComp-1B12M | ✗ | ✗ | 44.6 | 42.4 | 40.1 |
| DFN-5B12M | ✗ | ✗ | 49.9 | 48.5 | 43.5 |
| DataComp-1B12M | ✗ | ✓ | 51.9 | 71.8 | 47.8 |
| DFN-5B12M | ✗ | ✓ | 54.9 | 70.7 | 49.6 |
| DataComp-1B12M | ✓ | ✗ | 56.3 | 57.8 | 48.7 |
| DFN-5B12M | ✓ | ✗ | 59.5 | 60.4 | 50.0 |
| DataComp-1B12M | ✓ | ✓ | 61.6 | 72.8 | 53.7 |
| DFN-5B12M | ✓ | ✓ | 63.0 | 74.1 | 54.6 |

and `ViT-L-14-datacomp_xl_s13b_b90k` to result the best student performance. Here we investigate the effectiveness of DFN-pretrained models as teachers. DFN-pretrained CLIP models with ViT-L-14 and ViT-H-14 achieve the state-of-the-art performance on Avg. 38 evaluations of DataComp (Fang et al., 2024a) better than other popular models such as SigLIP (Zhai et al., 2023).

As the choice of the caption generator and the CLIP teachers may depend on each other, we reduce the complexity of our analysis by analyzing the effect of the CLIP teachers on synthetic captions from a CoCa model without fine-tuning (See Sec. 2.4). We explore the diversity of synthetic captions through fine-tuning in Sec. 2.5.

Table 3: **Optimal logit scale for distillation varies across teachers.** The dataset is DFN-5B12M with synthetic captions generated from CoCa-DFN-2B in Sec. 2.4. The loss coefficient $\lambda$ is set to 1.0 and trained using strong image augmentations.

| Teacher | Logit Scale | IN-val | Flickr30k | Avg. 38 |
|---|---|---|---|---|
| `datacomp_xl_s13b_b90k-CLIP-ViT-L-14` | 50 | 62.6 | 65.6 | 53.3 |
| `DFN2B-CLIP-ViT-L-14` | 70 | 65.5 | 68.0 | 56.5 |
| `DFN5B-CLIP-ViT-H-14` | 90 | 64.0 | 65.9 | 54.7 |
| `DFN5B-CLIP-ViT-H-14-384` | 55 | 64.6 | 67.6 | 54.4 |
| `DFN2B-CLIP-ViT-L-14-s39b` | 60 | 65.2 | 67.5 | 54.8 |

Table 4: **Ensemble of DFN CLIP teachers improve ImageNet-1k validation accuracy by 2.8%.** The dataset is DFN-5B12M with synthetic captions generated from CoCa-DFN-2B in Sec. 2.4. The loss coefficient $\lambda$ is set to 1.0 and trained using strong image augmentations. The optimal logit scales for each model is set independently based on Tab. 3.

| Teacher 1 | Teacher 2 | IN-val | Flickr30k | Avg. 38 |
|---|---|---|---|---|
| `ViT-L-14-openai` | `ViT-L-14-datacomp_xl_s13b_b90k` | 63.1 | 64.7 | 55.2 |
| `ViT-L-14-datacomp_xl_s13b_b90k` | `DFN5B-CLIP-ViT-H-14-384` | 64.5 | 67.8 | 54.5 |
| `ViT-L-14-datacomp_xl_s13b_b90k` | `DFN5B-CLIP-ViT-H-14` | 64.4 | 67.3 | 55.3 |
| `ViT-L-14-datacomp_xl_s13b_b90k` | `DFN2B-CLIP-ViT-L-14` | 65.3 | 68.1 | 56.2 |
| `DFN5B-CLIP-ViT-H-14-384` | `DFN5B-CLIP-ViT-H-14` | 64.7 | 66.9 | 54.9 |
| `DFN5B-CLIP-ViT-H-14-384` | `DFN2B-CLIP-ViT-L-14` | 65.8 | 68.6 | 56.2 |
| `DFN5B-CLIP-ViT-H-14` | `DFN2B-CLIP-ViT-L-14` | 65.2 | 68.0 | 55.8 |
| `DFN2B-CLIP-ViT-L-14-s39b` | `datacomp_xl_s13b_b90k` | 65.1 | 67.6 | 55.7 |
| `DFN2B-CLIP-ViT-L-14-s39b` | `DFN5B-CLIP-ViT-H-14-384` | 65.7 | 67.3 | 55.1 |
| `DFN2B-CLIP-ViT-L-14-s39b` | `DFN5B-CLIP-ViT-H-14` | 65.7 | 68.2 | 55.7 |
| `DFN2B-CLIP-ViT-L-14-s39b` | `DFN2B-CLIP-ViT-L-14` | 65.9 | 68.7 | 55.9 |

**Logit scaling.** CLIP models are trained with a logit scale that is tuned during the training in the range 0-100. MobileCLIP used the same logit scalar as the temperature scaling in the KD loss. We observe that

the logit scalar in DFN and DataComp models is not optimal for KD and tune that further. Table 3 shows the optimal logit scale used for each teacher to train a MobileCLIP-B model. We observe that the logit scale is not a sensitive hyperparameter where values within a range of 5 points achieve similar performance.

**Ensemble teachers.** We construct ensembles of size two using DataComp and DFN teachers. Table 4 shows the performance of training a MobileCLIP-B model using embeddings from various ensembles. We observe significant improvements compared with teachers used in MobileCLIP. Specifically, IN-val and Flickr30k improve by up to 3%. We choose the ensemble of `DFN2B-CLIP-ViT-L-14-s39b` and `DFN2B-CLIP-ViT-L-14` for MobileCLIP2 based on its performance and cost efficiency compared to other larger or higher resolution ensembles. We utilize the optimal logit scales for each member of the ensemble that is found independently. It is possible that the optimal logit scales for ensemble would vary when used together but we do not further optimize logit scales jointly.

### 2.4 DFN Caption Generators

Another source of reinforcements for training MobileCLIP2 is synthetic captions generated from an image caption generator. MobileCLIP used a single CoCa captioner which has a two-tower image-text architecture coupled with a text decoder (Yu et al., 2022). Compared with most recent VLMs, the text-decoder is fairly light-weight that results in an overall relatively faster caption generator compared with more recent VLMs (Liu et al., 2024b; Vasu et al., 2024a). As MobileCLIP generated multiple synthetic captions on billions of images, the cost of running CoCa was an important decision factor. They did not provide analysis on the choice of captioner but observed significant gains from training on synthetic captions compared with not using synthetic captions (7.4% for 30k training iterations). MobileCLIP generated 5 synthetic captions per image although they observed the majority of the gain comes from the first 1-2 synthetic captions.

We explore training a new CoCa model using the DFN dataset to improve the quality of synthetic captions. We adopt the same architecture as the CoCa model utilized in MobileCLIP based on the ViT-L/14 image encoder. They utilized the model trained on LAION-2B dataset and fine-tuned on MSCOCO-128k dataset. We pretrain the same architecture on DFN-2B for 13B seen samples using OpenCLIP (Ilharco et al., 2021).

Table 5: **Pretraining CoCa on DFN-2B without fine-tuning results in similar IN-1k performance but worse robustness and retrieval.** The dataset is DFN-5B12M, CLIP teachers are the same as MobileCLIP (OpenAI+DataComp-XL CLIP-ViT-L/14) and the architecture of the CoCa model is the same as CoCa-ViT-L/14. For distillation, the coefficient $\lambda$ is set to 1.0 (no CLIP loss) and use strong image augmentations. Values within one standard deviation of the best of each group are highlighted.

| Distill. | High Aug. | CoCa | | IN-val | Flickr30k | Avg. 38 |
|---|---|---|---|---|---|---|
| | | LAION-2B → MSCOCO-128k | DFN-2B | | | |
| ✗ | ✗ | ✗ | ✗ | 49.9 | 48.5 | 43.5 |
| ✗ | ✗ | ✓ | ✗ | 54.9 | 70.6 | 49.6 |
| ✗ | ✓ | ✓ | ✗ | 51.1 | 65.7 | 45.3 |
| ✗ | ✓ | ✗ | ✓ | 54.6 | 55.1 | 46.2 |
| ✗ | ✓ | ✓ | ✓ | 56.8 | 67.2 | 48.4 |
| ✓ | ✓ | ✗ | ✗ | 59.5 | 60.3 | 50.0 |
| ✓ | ✓ | ✓ | ✗ | 63.0 | 74.1 | 54.6 |
| ✓ | ✓ | ✗ | ✓ | 63.1 | 64.7 | 55.2 |
| ✓ | ✓ | ✓ | ✓ | 63.4 | 72.0 | 55.1 |

Table 5 demonstrates the impact of DFN-CoCa synthetic captions on the performance with and without distillation. We observe that utilizing DFN-CoCa synthetic captions results in improved IN-val and Avg. 38 performance but negatively impacts retrieval. As we observe in Sec. 2.5, the retrieval performance recovers with fine-tuning on high-quality datasets such as MSCOCO. We further observe the synthetic captions from the original CoCa model can be used together with DFN-CoCa captions to provide additional gains but these gains are small with distillation.

## 2.5 Fine-tuning Caption Generators

In Sec. 2.4, we showed that pretraining a CoCa model on DFN-2B results in improved IN-val and Avg. 38 performance when utilized for multi-modal reinforced training. However, the retrieval performance falls behind which is due to the lack of fine-tuning on a high-quality dataset. MobileCLIP used a CoCa model fine-tuned on MSCOCO (Chen et al., 2015). MSCOCO-2017 contains 123k images with captions that have higher quality compared to average image-text pairs in DataComp and DFN datasets.

In this section, we study the impact of fine-tuning on various high-quality datasets. In addition to 123k samples from MSCOCO which we refer to as MSCOCO-123k, we also consider a subset of 38k samples with permissive licenses (CC Attribution 2.0, CC Attribution-ShareAlike 2.0, and CC Attribution-NoDerivs 2.0) which we refer to as MSCOCO-38k. We also consider GBC-1M/10M (Hsieh et al., 2024), DOCCI-9k-short/extended/complete (Onoe et al., 2025), DCI-8k (Urbanek et al., 2024), and ReCap-COCO-30k (Li et al., 2024). We fine-tune DFN-CoCa on each dataset for 12M seen samples using the same loss as CoCa pretraining.

Table 6: The dataset is DFN-5B12M, CLIP teachers are our selected DFN models (DFN2B-CLIP-ViT-L-14-s39b and DFN2B-CLIP-ViT-L-14) and the architecture of the CoCa model is the same as CoCa-ViT-L/14. For distillation, the coefficient is set to 1.0 (no CLIP loss) and use strong image augmentations.

| Base Dataset | FT Dataset | Context len. | IN-val | Flickr30k | Avg. 38 |
|---|---|---|---|---|---|
| LAION-2B | MSCOCO-123k | 77 | $65.4_{0.4}$ | $75.8_{0.3}$ | $56.2_{0.6}$ |
| DFN-2B | - | 77 | 65.9 | 68.7 | 55.9 |
| DFN-2B | MSCOCO-123k | 77 | 65.9 | 76.0 | 56.2 |
| DFN-2B | MSCOCO-38k | 77 | $65.9_{0.3}$ | $75.4_{0.2}$ | $56.5_{0.3}$ |
| DFN-2B | GBC1M-short | 77 | 65.8 | 75.0 | 56.6 |
| DFN-2B | DOCCI | 77 | 66.3 | 72.6 | 57.3 |
| DFN-2B | DCI-short | 77 | 65.9 | 74.0 | 56.3 |
| DFN-2B | DCI-extended | 77 | 65.7 | 73.5 | 56.1 |
| DFN-2B | DCI-complete | 77 | 65.8 | 73.8 | 56.2 |
| DFN-2B | Recap-COCO-30K | 77 | 65.1 | 73.5 | 55.5 |
| DFN-2B | GBC-1M-long | 255 | 64.7 | 72.4 | 55.1 |
| DFN-2B | GBC-10M-short-relation | 255 | 65.2 | 73.8 | 55.4 |
| DFN-2B | GBC-10M-long | 255 | 64.6 | 71.9 | 54.6 |
| DFN-2B | DOCCI | 255 | 66.1 | 74.0 | 57.2 |
| DFN-2B | DCI-extended | 255 | 65.7 | 75.1 | 55.9 |
| DFN-2B | DCI-complete | 255 | 65.6 | 74.0 | 56.8 |
| DFN-2B | 5×2 | 77 | $65.9_{0.2}$ | $74.7_{0.4}$ | $56.3_{0.2}$ |
| DFN-2B | 10×1 | 77 | $66.0_{0.1}$ | $75.1_{0.6}$ | $56.5_{0.3}$ |

**Fine-tuning on MSCOCO38k and MSCOCO128k**. We observe that restricting fine-tuning to MSCOCO samples with permissive licenses does not have a negative impact on performance.

**Ablation on number of synthetic captions and beam search**. (Vasu et al., 2024c) observed that even though one can generate multiple synthetic captions from a CoCa model, their effectiveness saturates at 2 per sample for classification tasks. We observe similar results using a single CoCa model with various sampling strategies. We explore varying the generation method and hyperparameters. Specifically, we used top-p, top-k, and beam-search and observed that beam-search results in qualitatively more diverse captions, however, we did not observe any improvement in downstream performance when utilized for reinforced training.

**Fine-tuning on GBC1M, GBC12M, DOCCI, DCI, ReCap-COCO30k**. We observe that most fine-tuning datasets underperform MSCOCO fine-tuning or perform on-par within one standard deviation. An exception is fine-tuning on DOCCI results in 0.8% improvement in average of 38 evaluations which is more than one standard deviation from the MSCOCO-38k results.

**Effect of context length.** The context length for training CLIP and CoCa models is typically set to 77. We explore training CoCa models to generate longer captions by setting the context length for training and generation to 255. Most results stay within one standard deviation. Recent works have improved the support for long captions in CLIP models with improved loss functions and training strategies (Zhang et al., 2024; Zheng et al., 2024; Najdenkoska et al., 2024). We leave extending these modifications to CoCa models for future work.

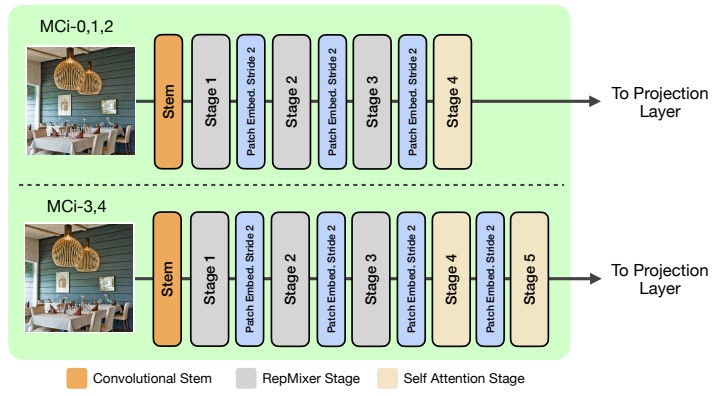
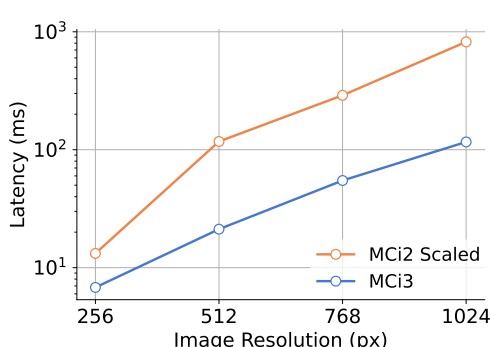

(a) **MCi architectures for new larger variants**. For smaller variants, the image encoder has four distinct stages of compute and for larger variants, we use five stages. The projection layer for MCi models include a Global Avg. Pooling layer followed by a linear layer.

(b) **5-Stage achieves lower latency at higher resolutions.** Both models MCi2-Scaled and MCi3 are of the same size, with MCi3 using a 5-stage design.

Figure 3: **MobileCLIP2 architecture and latency.**

**Effect of synthetic caption diversity**. We further explore training with a diverse collection of captions generated from an ensemble of CoCa models fine-tuned on different datasets. The motivation is the diversity in fine-tuning datasets would increase the divresity in synthetic captions and hence an increase in the effectiveness of additional synthetic captions. We observe that utilizing up to 10 different CoCa models results in a performance that is still within one standard deviation of the best performance.

**Reinforced DFN datasets.** Our final datasets small DFNDR-5B12M and DFNDR-2B12M consist of 5 synthetic captions with MSCOCO-38k fine-tuning, and embeddings from the ensemble of two DFN2B-ViT-L/14 teachers discussed in Sec. 2.3 for 30 image augmentations as well as ground-truth and synthetic captions. We explored training on only the 2B subset of DFN versus the full 5B set that was expanded with 3B samples outside of the 12B pool of DataComp. Tab. 7 shows that the average performance on 38 evaluations is within standard deviation for both datasets while ImageNet-1k validation accuracy is better with the 12M samples from 5B. However, we did not observe the improvement to hold when training at larger scales and restricted our recipe to the 2B dataset.

Table 7: **DFNDR-5B12M and DFNDR-2B12M perform similarly on average 38 evaluations.**

| Dataset | IN-val | Flickr30k | Avg. 38 |
|---|---|---|---|
| DFNDR-5B12M | $65.9_{0.3}$ | $75.4_{0.2}$ | $56.5_{0.3}$ |
| DFNDR-2B12M | 65.5 | 74.8 | 56.4 |

# 3  Architecture

Our MobileCLIP2 consists of similar architectures to MobileCLIP as well as two new variants. Specifically, we train MobileCLIP2-S0, MobileCLIP2-S2, and MobileCLIP2-B where we utilize the standard "Base" text encoder for MobileCLIP2-S0 and drop the S1 variant. In addition to architectures introduced in MobileCLIP, we introduce two new variants in MobileCLIP2 family, i.e., MobileCLIP2-S3 and MobileCLIP2-S4. The text encoders for these variants are pure transformer-based architectures and the image encoders are based on FastViT (Vasu et al., 2023b), which uses train-time overparameterization blocks introduced in (Vasu et al., 2023a). The smaller variants, MCi0, MCi1, and MCi2 are hybrid vision transformers with four distinct stages of compute. We introduce an additional transformer stage for MCi3 and MCi4 preceded by $4\times$ down-sampling of input tensor as shown in Fig. 3a. The 5-stage design has two advantages when scaled up; First, the

parameters can be distributed across five stages with the largest layers operating on four times fewer tokens. Second, the design scales more effectively to higher resolutions.

We empirically validate our design choices across various image resolutions. In Fig. 3b, we scale MCi2 to match the size of MCi3 (125 million parameters) and benchmark its performance across four input resolutions. Our results show that MCi3, with its five-stage design, offers a significantly better trade-off compared to a scaled MCi2. At low image resolution, i.e., 256×256, MCi3 is 1.9× faster than similar sized MCi2 and for larger input resolutions, i.e., as 1024×1024, MCi3 is 7.1× faster than a similar sized MCi2. Responsiveness at higher resolutions is particularly important when the image encoder is fine-tuned for dense prediction tasks such as image segmentation, where the input image resolution is 512×512.

## 4 Experiments

In this section, we train a new family of efficient CLIP models, MobileCLIP2, and evaluate on a diverse set of tasks. Following our findings in Sec. 2, we create the reinforced dataset, DFNDR-2B, which contains five synthetic captions generated from our CoCa-ViT-L/14 model pretrained on DFN-2B and fine-tuned on MSCOCO-38K. DFNDR-2B also contains image-text embeddings from an ensemble of CLIP models, `DFN2B-CLIP-ViT-L-14-s39b` and `DFN2B-CLIP-ViT-L-14`, for all images, ground-truth captions, and synthetic captions. We train a more diverse family of architectures compared with MobileCLIP and evaluate their performance on 38 zero-shot classification tasks (Gadre et al., 2023). Particularly, we introduce MobileCLIP2-S3 and MobileCLIP2-S4 architectures trained on DFNDR-2B as well as variants trained on DataCompDR-1B which we refer to as MobileCLIP-S3 and MobileCLIP-S4. Table 8 shows our results compared with other models with similar latencies. Details of training and hyperparameters are described in Appx. A.

We compare MobileCLIP2 to prior small CLIP architectures TinyCLIP (Wu et al., 2023) trained on LAION (Schuhmann et al., 2022; 2021) and ACED (Udandarao et al., 2024). We also compare with larger models from OpenAI's CLIP (Radford et al., 2021), DataComp (Gadre et al., 2023), VeCLIP (Lai et al., 2023), EVA (Sun et al., 2023), DFN (Fang et al., 2024a), SigLIP (Zhai et al., 2023), and SigLIP2 (Tschannen et al., 2025). We evaluate all models using OpenCLIP (Ilharco et al., 2021) and DataComp (Gadre et al., 2023). In some cases such as SigLIP2, we observe positive/negative gaps with reported results in their paper.

MobileCLIP2 achieves state-of-the-art ImageNet-1k validation zero-shot accuracies at various latencies. Notably, MobileCLIP2-S4 matches the zero-shot accuracy of SigLIP-SO400M/14 on ImageNet validation set while being 2× smaller and improves on DFN ViT-L/14 at 2.5× lower latency. We also improve on ImageNet-1k performance of ACED models considering their latencies. As ACED optimized their models for low inference flops, the latency of both ACED-F1 and ACED-F2 are comparable to our MobileCLIP2-S2 architecture while still have higher latency and more parameters. SigLIP-B/16 and SigLIP2-B/16 models are more comparable in size and latency to our new larger architectures. Particularly, SigLIP2 models have substantially larger text-encoders compared to SigLIP models.

We note that our models pretrained on DFNDR-2B do not always achieve state-of-the-art retrieval performance. We attribute this to the bias of DFNDR-2B dataset towards zero-shot classification tasks and particularly ImageNet-1k. We observe that models trained on DataComp, WebLI, and their derivatives may achieve higher retrieval performance compared to DFN datasets and derivatives while lower on Avg. 38 performance. As such, we also train our new architectures on DataCompDR-1B referred to as MobileCLIP-S3 and MobileCLIP-S4. The combination of these two families of architectures will provide flexibility for broader applications.

### 4.1 VLM evaluations

We report vision-language evaluations using MobileCLIP2 pretrained models in the LLaVA-1.5 setup (Liu et al., 2024a). We keep the vision backbone frozen for all the runs and use Qwen2-7B instead of Vicuna-7B. All other training details are the same as the original LLaVA-1.5 setup, more details are provided in appendix. We evaluate ViT-B/16 models pretrained on DataComp, DFN, DataCompDR, and DFNDR for 13B seen samples. In Tab. 9 we observe that on average training on DFNDR achieves 3.5% higher accuracy compared with DFN pretrained model, 1.6% better than DataComp pretrained model, and 0.6% better than DataCompDR pretrained model.

Table 8: **MobileCLIP2 family of models has the best average performance at various latencies.** Retrieval performances are reported @1. Last column shows average performance on 38 datasets as in OpenCLIP (Ilharco et al., 2021). Models are grouped by their total latency in increasing order and by performance within each group. "Base" refers to standard CLIP Transformer-based (Vaswani et al., 2017) text encoder with 12 layers, and "Custom" stands for customized text encoder used in the respective method. Models with substantially higher latencies and/or larger model sizes are grayed out.

| Name | Dataset | Seen Samples | Resolution | Image Encoder | Text Encoder | Params (M) (img+txt) | Latency (ms) (img+txt) | Zero-shot CLS IN-val | IN-shift | Flickr30k Ret. T→I | I→T | COCO Ret. T→I | I→T | Avg. Perf. on 38 |
|---|---|---|---|---|---|---|---|---|---|---|---|---|---|---|
| TinyCLIP-RN19M | LAION-400M | 15.2B | 224 | ResNet-19M | Custom | 18.6 + 44.8 | 1.9 + 1.9 | 56.3 | 43.6 | 58.0 | 75.4 | 30.9 | 47.8 | 48.3 |
| TinyCLIP-RN30M | LAION-400M | 15.2B | 224 | ResNet-30M | Custom | 29.6 + 54.2 | 2.6 + 2.6 | 59.1 | 45.7 | 61.5 | 80.1 | 33.8 | 51.6 | 50.2 |
| TinyCLIP-40M/32 | LAION-400M | 15.2B | 224 | ViT-40M/32 | Custom | 39.7 + 44.5 | 3.0 + 1.9 | 59.8 | 46.5 | 59.1 | 76.1 | 33.5 | 48.7 | 51.2 |
| MobileCLIP-S0 | DataCompDR-1B | 13B | 256 | MCi0 | MCt | 11.4 + 42.4 | 1.5 + 1.6 | 67.8 | 55.1 | 67.7 | 85.9 | 40.4 | 58.7 | 58.1 |
| ACED-F0 | DataComp-1B | 13B | 256 | ViT-S/32 | Small | 22.7 + 28.8 | 2.1 + 1.8 | 68.5 | (-) | 71.4 | 87.6 | 41.2 | 60.8 | (-) |
| **MobileCLIP2-S0** | DFNDR-2B | 13B | 256 | MCi0 | Base | 11.4 + 63.4 | 1.5 + 3.3 | 71.5 | 57.6 | 69.2 | 86.6 | 43.7 | 62.7 | **59.7** |
| OpenAI-RN50 | OpenAI-400M | 13B | 224 | ResNet-50 | Base | 38.3 + 63.4 | 3.3 + 3.3 | 59.8 | 45.1 | 57.4 | 80.0 | 28.5 | 48.8 | 48.1 |
| TinyCLIP-61M/32 | LAION-400M | 15.2B | 224 | ViT-61M/32 | Base | 61.4 + 54.0 | 4.3 + 2.6 | 62.4 | 48.7 | 62.6 | 78.7 | 36.5 | 52.8 | 53.0 |
| TinyCLIP-63M/32 | LAION-400M YFCC-15M | 15.8B | 224 | ViT-63M/32 | Custom | (-) | (-) | 64.5 | (-) | 66.0 | 84.9 | 38.5 | 56.9 | (-) |
| MobileCLIP-S1 | DataCompDR-1B | 13B | 256 | MCi1 | Base | 21.5 + 63.4 | 2.5 + 3.3 | 72.6 | 60.7 | 71.0 | 89.2 | 44.0 | 62.2 | **61.3** |
| OpenAI-RN101 | OpenAI-400M | 13B | 224 | ResNet-101 | Base | 56.3 + 63.4 | 4.3 + 3.3 | 62.3 | 48.5 | 58.0 | 79.0 | 30.7 | 49.8 | 50.3 |
| OpenAI-B/32 | OpenAI-400M | 13B | 224 | | | 63.3 + 63.4 | | 63.3 | 48.5 | 58.8 | 78.9 | 30.4 | 50.1 | 52.5 |
| LAION-B/32 | LAION-2B | 32B | 224 | ViT-B/32 | Base | 86.2 + 63.4 | 5.9 + 3.3 | 65.7 | 51.9 | 66.4 | 84.4 | 39.1 | 56.2 | 54.8 |
| DataComp-B/32 | DataComp-1B | 13B | 224 | | | | | 69.2 | 55.2 | 61.1 | 79.0 | 37.1 | 53.5 | 58.0 |
| DataComp-B/32-256 | DataComp-1B | 34B | 256 | ViT-B/32 | Base | 86.2 + 63.4 | 6.2 + 3.3 | 72.8 | 58.7 | 64.9 | 84.8 | 39.9 | 57.9 | 60.9 |
| SigLIP-B/32 | WebLI-10B | 40B | 256 | ViT-B/32 | Custom | 94.6 + 282.3 | 6.3 + 6.3 | 73.8 | 57.8 | 73.2 | 88.0 | 47.9 | 64.9 | 61.9 |
| MobileCLIP-S2 | DataCompDR-1B | 13B | 256 | MCi2 | Base | 35.7 + 63.4 | 3.6 + 3.3 | 74.4 | 63.1 | 73.4 | 90.3 | 45.4 | 63.4 | 63.7 |
| ACED-F1 | DataComp-1B | 13B | 256 | ViT-B/32 | Small | 86.2 + 28.8 | 6.2 + 1.8 | 74.9 | (-) | 77.9 | 90.3 | 47.3 | 74.9 | (-) |
| ACED-F2 | DataComp-1B | 13B | 256 | ViT-B/24 | Small | 86.2 + 28.8 | 6.5 + 1.8 | 76.9 | (-) | 79.5 | 91.1 | 49.7 | 66.9 | (-) |
| **MobileCLIP2-S2** | DFNDR-2B | 13B | 256 | MCi2 | Base | 35.7 + 63.4 | 3.6 + 3.3 | 77.2 | 64.7 | 74.8 | 90.4 | 48.8 | 66.7 | **64.1** |
| VeCLIP-B/16 | WIT-200M | 6.4B | 224 | | Base | 86.2 + 63.4 | 11.5 + 3.3 | 64.6 | (-) | 76.3 | 91.1 | 48.4 | 67.2 | (-) |
| OpenAI-B/16 | WIT-400M | 13B | 224 | | Base | 86.2 + 63.4 | 11.5 + 3.3 | 68.3 | 55.9 | 67.7 | 85.9 | 40.4 | 58.7 | 58.1 |
| LAION-B/16 | LAION-2B | 34B | 224 | | Base | 86.2 + 63.4 | 11.5 + 3.3 | 70.2 | 56.6 | 69.8 | 86.3 | 42.3 | 59.4 | 58.7 |
| EVA02-B/16 | Merged-2B | 8B | 224 | | Base | 86.2 + 63.4 | (-) | 74.7 | 59.6 | 71.5 | 86.0 | 42.2 | 58.7 | 58.9 |
| DFN-B/16 | DFN-2B | 13B | 224 | ViT-B/16 | Base | 86.2 + 63.4 | 11.5 + 3.3 | 76.2 | 62.3 | 69.1 | 85.4 | 43.4 | 60.4 | 60.9 |
| DataComp-B/16 | DataComp-1B | 13B | 224 | | Base | 86.2 + 63.4 | 11.5 + 3.3 | 73.5 | 60.8 | 69.8 | 86.3 | 42.3 | 59.4 | 61.5 |
| MobileCLIP-B | DataCompDR-1B | 13B | 224 | | Base | 86.3 + 63.4 | 10.4 + 3.3 | 76.8 | 65.6 | 77.3 | 91.4 | 50.6 | 68.8 | 65.2 |
| MobileCLIP-B (LT) | DataCompDR-1B | 39B | 224 | | Base | 86.3 + 63.4 | 10.4 + 3.3 | 77.2 | 66.1 | 76.9 | 92.3 | 50.0 | 68.7 | 65.8 |
| **MobileCLIP2-B** | DFNDR-2B | 13B | 224 | | Base | 86.3 + 63.4 | 10.4 + 3.3 | 79.4 | 66.4 | 76.5 | 89.7 | 49.9 | 67.5 | **65.8** |
| SigLIP-B/16 | WebLI | 40B | 224 | ViT-B/16 | Custom | 92.9 + 110.3 | 9.9 + 5.8 | 76.0 | 61.0 | 74.7 | 89.1 | 47.8 | 65.7 | 62.3 |
| SigLIP-B/16-256 | WebLI | 40B | 256 | ViT-B/16 | Custom | 92.9 + 110.3 | 11.4 + 5.8 | 76.5 | 62.0 | 75.0 | 90.4 | 48.4 | 66.1 | 62.3 |
| SigLIP-B/16 | WebLI-10B | 40B | 224 | ViT-B/16 | Custom | 92.9 + 282.3 | 9.9 + 6.3 | 78.5 | 63.9 | 79.3 | 93.1 | 53.2 | 69.4 | 64.6 |
| SigLIP-B/16-256 | WebLI-10B | 40B | 256 | ViT-B/16 | Custom | 92.9 + 282.3 | 11.4 + 6.3 | 79.3 | 65.3 | 80.2 | 93.2 | 54.1 | 70.8 | 64.6 |
| **MobileCLIP-S3** | DataCompDR-1B | 13B | 256 | MCi3 | Large | 125.1 + 123.6 | 8.0 + 6.6 | 78.3 | 68.2 | 77.9 | 93.1 | 51.3 | 68.8 | 66.3 |
| **MobileCLIP2-S3** | DFNDR-2B | 13B | 256 | MCi3 | Large | 125.1 + 123.6 | 8.0 + 6.6 | 80.7 | 68.9 | 77.3 | 91.6 | 50.9 | 68.4 | **66.8** |
| SigLIP-L/16 | WebLI | 40B | 256 | ViT-L/16 | Custom | 316.0 + 336.2 | 38.2 + 19.1 | 80.4 | 66.6 | 79.0 | 91.8 | 52.3 | 70.8 | 65.6 |
| DFN-L/14-quickgelu | DFN-2B | 13B | 224 | ViT-L/14 | Large | 304.3 + 123.6 | 57.9 + 6.6 | 81.4 | 68.8 | 78.5 | 89.0 | 53.7 | 66.8 | 66.9 |
| **MobileCLIP-L/14** | DataCompDR-1B | 13B | 224 | ViT-L/14 | Large | 304.3 + 123.6 | 57.9 + 6.6 | 79.5 | 69.9 | 75.3 | 91.3 | 47.6 | 66.5 | 66.9 |
| **MobileCLIP2-S4** | DFNDR-2B | 13B | 256 | MCi4 | Large | 321.6 + 123.6 | 19.6 + 6.6 | 81.9 | 70.3 | 78.0 | 92.4 | 51.5 | 69.3 | **67.5** |
| **MobileCLIP2-L/14** | DFNDR-2B | 13B | 224 | ViT-L/14 | Large | 304.3 + 123.6 | 57.9 + 6.6 | 81.9 | 70.2 | 77.2 | 92.0 | 51.6 | 69.0 | **67.8** |
| **MobileCLIP-S4** | DataCompDR-1B | 13B | 256 | MCi4 | Large | 321.6 + 123.6 | 19.6 + 6.6 | 79.4 | 69.7 | 79.5 | 94.9 | 52.1 | 70.3 | **68.1** |
| SigLIP-SO400M/14 | WebLI | 40B | 224 | So-400M | Custom | 427.7 + 449.7 | (-) | 82.0 | 69.5 | 75.2 | 91.0 | 51.8 | 69.7 | 68.1 |
| SigLIP2-L/16 | WebLI-10B | 40B | 256 | ViT-L/16 | Custom | 316.0 + 565.6 | 38.2 + 19.8 | 82.3 | 70.5 | 81.8 | 94.6 | 54.7 | 72.0 | 68.3 |
| SigLIP2-SO400M/14 | WebLI-10B | 40B | 224 | So-400M | Custom | 427.7 + 707.8 | (-) | 83.2 | 72.0 | 82.8 | 93.9 | 55.5 | 71.9 | 69.1 |

Table 9: **VLM evaluations in LLaVA-1.5 setup.** ViT-B/16 pretrained models reach **3.5% higher accuracy** compared with DFN pretrained model, 1.6% better than DataComp pretrained model, and 0.6% better than DataCompDR pretrained model.

| Dataset | GQA | SQA | TextVQA | POPE | MMMU | MMB | VizWiz | VQAv2 | Avg. |
|---|---|---|---|---|---|---|---|---|---|
| DataComp-1B | 59.6 | 71.5 | **50.5** | 81.8 | 42.6 | 59.1 | 51.8 | 70.7 | 61.0 |
| DFN-2B | 56.9 | 71.3 | 46.0 | 81.4 | 41.9 | 52.2 | **56.1** | 66.9 | 59.1 |
| DataCompDR-1B | 60.3 | **73.1** | 50.4 | 81.7 | 43.6 | 60.2 | 54.9 | 72.1 | 62.0 |
| DFNDR-2B | **60.4** | 72.9 | 49.9 | **83.3** | **45.2** | **61.9** | 54.5 | **72.4** | **62.6** |

## 4.2 Dense Prediction tasks

We evaluate the quality of the visual representations learned by finetuning the image encoder on dense prediction tasks like object detection, semantic segmentation and depth estimation. In Table 10, we report performance of ViT-B/16 model with MaskRCNN He et al. (2017) head for instance segmentation on MS-COCO Chen et al. (2015) dataset. All models were trained using MMDetection library Chen et al. (2019) using 1× schedule with single scale testing as described in Wei et al. (2023). We follow finetuning setup described in Wei et al. (2023), more details in appendix. In Table Table 11, we report performance of ViT-B/16 model with UperNet Xiao et al. (2018) head, trained using the same setup described in Liu et al. (2024c) on ADE20k (Zhou et al., 2017) dataset. In Table 12, we report Root Mean Square Error (RMSE) on NYUv2 dataset Nathan Silberman & Fergus (2012). We use the same settings as described in Vasu et al. (2024b), more details are provided in appendix.

Table 12: Results on NYUv2 for depth estimation following the same settings as Wei et al. (2023). All results are for ViT-B/16 models.

| Method | Dataset | RMSE($\downarrow$) |
|---|---|---|
| CatLIP Mehta et al. (2024) | DataComp | 0.394 |
| MAE He et al. (2022) | IN-1K | 0.383 |
| MAWS Singh et al. (2023) | IG-3B | 0.371 |
| FD-CLIP Wei et al. (2023) | OpenAI-WIT + IN-1K | 0.352 |
| MAE Singh et al. (2023) | IG-3B | **0.348** |
| CLIP Radford et al. (2021) | OpenAI-WIT | 0.416 |
| **MobileCLIP2** | DFNDR-2B | 0.356 |

Table 13: Comparison pretraining methods for semantic segmentation on ADE-20k. For reference, we have included recent state-of-the-art semantic segmentation models (in gray).

| Encoder | Decoder | Pre-Training | Resolution | # Params(M) | mIoU |
|---|---|---|---|---|---|
| InternImage-B Wang et al. (2023) | UperNet Xiao et al. (2018) | Sup. IN-1K | 512×512 | 128.0 | 50.8 |
| ViT-Adapter-B Chen et al. (2023) | SemanticFPN Kirillov et al. (2019) | Sup. IN-22K | 512×512 | 104.6 | 50.7 |
| ViT-Adapter-B Chen et al. (2023) | UperNet Xiao et al. (2018) | Sup. IN-22K | 512×512 | 133.9 | 51.9 |
| Swin-L Liu et al. (2021) | UperNet Xiao et al. (2018) | Sup. IN-22K | 640×640 | 234.1 | 52.1 |
| MCi0 | SemanticFPN Kirillov et al. (2019) | Sup. IN-1K | 512×512 | 14.5 | 44.8 |
| MCi2 | SemanticFPN Kirillov et al. (2019) | Sup. IN-1K | 512×512 | 38.5 | 48.9 |
| MCi0 | SemanticFPN Kirillov et al. (2019) | **MobileCLIP2** | 512×512 | 14.5 | 47.0 (+2.2) |
| MCi2 | SemanticFPN Kirillov et al. (2019) | **MobileCLIP2** | 512×512 | 38.5 | 51.6 (+2.7) |

Additionally, we assess the performance of smaller MobileCLIP2 variants on dense prediction tasks. Popular pretraining methods like MAE (He et al., 2022), are not directly applicable to hierarchical convolutional and hybrid architectures such as our MCi models, hence we compare MobileCLIP2 pretraining with supervised pretraining for the same architectures. In Tabs. 13 and 14, we see that MobileCLIP2 pretraining is significantly better than supervised pretraining and can serve as a good pretraining choice for hierarchical architectures.

Table 10: Object detection and instance segmentation results on MS-COCO with Mask-RCNN head trained for 1× schedule. All models are ViT-B/16.

| Method | Dataset | mAP$^{box}$ | mAP$^{mask}$ |
|---|---|---|---|
| CatLIP Mehta et al. (2024) | DataComp | 45.7 | 40.6 |
| MAE He et al. (2022) | IN-1K | 46.5 | 40.9 |
| MAE Singh et al. (2023) | IG-3B | 46.4 | 42.1 |
| MAWS Singh et al. (2023) | IG-3B | 48.0 | **43.4** |
| FD-CLIP Wei et al. (2023) | OpenAI-WIT + IN-1K | **48.2** | 42.5 |
| CLIP Radford et al. (2021) | OpenAI-WIT | 45.0 | 39.8 |
| **MobileCLIP2** | DFNDR-2B | 47.0 | 41.8 |

Table 11: Semantic segmentation results on ADE20k using UperNet decoder. All models are ViT-B/16.

| Method | Dataset | mIoU | mAcc |
|---|---|---|---|
| MAE He et al. (2022) | IN-1K | 48.1 | 58.9 |
| dBOT Liu et al. (2024c) | IN-1K | 49.5 | 60.7 |
| MAWS Singh et al. (2023) | IG-3B | 50.4 | 61.5 |
| CatLIP Mehta et al. (2024) | DataComp | 50.6 | 61.8 |
| FD-CLIP Wei et al. (2023) | OpenAI-WIT + IN-1K | 51.7 | - |
| CLIP Radford et al. (2021) | OpenAI-WIT | 49.5 | - |
| **MobileCLIP2** | DFNDR-2B | **52.8** | **64.0** |

# 5 Related Work

Improving the training of multi-modal models focus on three aspects: data, objective function and architecture. Our MobileCLIP2 builds on MobileCLIP and provides improvements in all three aspects.

Data approaches either filter a dataset or augment it with additional information. Basic filtering methods begin by selecting or crawling a large dataset of candidate image-text pairs and filter using ad-hoc rules based on the URLs or statics of the images and captions (Radford et al., 2021; Schuhmann et al., 2021; 2022; Xu et al., 2024). More advanced filtering methods involve filtering models trained on high-quality data utilized to remove low-quality image-text pairs. These methods may utilize a pretrained CLIP model (Gadre et al., 2023) or more specialized filtering models (Fang et al., 2024a). The challenge with data methods is that the biases introduced by ad-hoc rules or pretrained models. For example, most publicly available datasets such as DataComp are filtered for English-only data which limits the capabilities of models on non-English tasks (Carlsson et al., 2022; Nguyen et al., 2024; Pouget et al., 2024). Alternatively, pretrained models may be used for active data selection based on the sample difficulty (Evans et al., 2024a;b). It has also been observed that repeating high-quality data achieves higher utilization (Goyal et al., 2024).

Table 14: Comparison pretraining methods for object detection task on MS-COCO using MaskRCNN He et al. (2017) detection head. All models are trained for 1× schedule. For reference we have included recent state-of-the-art object detection models (in gray).

| Model | Pre-Training | # Params(M) | mAP$^{box}$ | mAP$^{mask}$ |
|---|---|---|---|---|
| ViT-Adapter-B Chen et al. (2023) | Sup. IN-1K | 284 | 47.0 | 41.8 |
| InternImage-B Wang et al. (2023) | Sup. IN-1K | 115 | 48.8 | 44.0 |
| ViT-Adapter-L Chen et al. (2023) | Sup. IN-22K | 347.9 | 48.7 | 43.3 |
| MCi0 | Sup. IN-1K | 31.0 | 41.8 | 38.0 |
| MCi2 | Sup. IN-1K | 55.0 | 46.6 | 41.7 |
| MCi0 | **MobileCLIP2** | 31.0 | 44.4 (+2.6) | 39.6 (+1.6) |
| MCi2 | **MobileCLIP2** | 55.0 | 49.1 (+2.5) | 43.2 (+1.5) |

More broadly, the output of pretrained models can be stored as part of a new augmented dataset. For example, various works utilize image-captioning models to generate synthetic captions for images in a dataset (Yang et al., 2023a; Nguyen et al., 2023; Lai et al., 2023; Liu et al., 2024d; Li et al., 2024). Large language models can also be used to rewrite ground-truth captions (Fan et al., 2023) as well as together with text-to-image models to generate fully synthetic datasets (Hammoud et al., 2024). MobileCLIP introduced the multi-modal dataset reinforcement where they utilized an image-caption model to generate synthetic captions as well as an ensemble of large CLIP models to store CLIP embeddings for multiple image augmentations and synthetic captions and store them efficiently (Vasu et al., 2024c). We follow a similar approach while improving both the caption generator and CLIP embedding generators through better DFN models (Fang et al., 2024a).

Another approach is to improve the objective function of multi-modal training. The original CLIP paper utilized a contrastive loss that encourages the representations of images and texts paired in the dataset to be kept close to each other while staying farther away from other images and texts in a mini-batch (Radford et al., 2021). SigLIP introduced a variant based on Sigmoid instead of Softmax that achieves higher training efficiency at larger batch sizes (Zhai et al., 2023; Tschannen et al., 2025). Other methods utilize objectives based on image masking (Yang et al., 2023b; Fang et al., 2023; Sun et al., 2023; Li et al., 2023b) and unimodal self-supervision (Mu et al., 2022; Li et al., 2021) as well as multi-resolution training (Li et al., 2023a) for cost-effective training. Multi-modal distillation achieves more significant improvements, particularly for smaller architecture variants (Wang et al., 2022b; Kuang et al., 2023; Wang et al., 2022a; Wu et al., 2023). Notably, MobileCLIP (Vasu et al., 2024c) achieved high training efficiency by utilizing an offline knowledge distillation method (Shen & Xing, 2022; Yun et al., 2021; Faghri et al., 2023). We utilize a similar objective function as MobileCLIP that includes embedding distillation on image-text pairs and synthetic captions.

Lastly, architectural improvements seek improved inference efficiency and higher performance given a parameter, flops, or latency budget. CLIP architectures are often borrowed from uni-modal image and text models. Particularly, the original CLIP and various followup works utilized standard ViT architectures together with a modified BERT text encoder (Dosovitskiy et al., 2020; Devlin et al., 2019; Radford et al., 2021). Efficient architectures for CLIP include TinyCLIP that prunes ViT (Wu et al., 2023), Cao et al. (2023) that reduce tokens, and Evans et al. (2024b) that reduce the parameters for lower flops. MobileCLIP introduced efficient architectures specifically design for CLIP where they introduced a low latency convolution-transformer hybrid architectures for both their image and text encoders. We further improve on their architectures by introducing two new variants that fill the large latency gap between common B and L architectures.

## 6 Conclusion

We introduce MobileCLIP2, a new family of low latency image-text models, achieving state-of-the-art ImageNet-1k zero-shot validation accuracy. Our methodology improves multi-modal reinforced training by utilizing stronger CLIP teachers as well as our newly trained image-captioning models. We particularly perform a comprehensive study of tuning and ensembling CLIP teachers as well as training and fine-tuning efficient image-captioning models. Notably, MobileCLIP2-S4 matches the zero-shot accuracy of SigLIP-SO400M/14 on ImageNet-1k while being 2× smaller and improves on DFN ViT-L/14 at 2.5× lower latency. We release our model checkpoints and data generation code that facilitates dataset generation at scale.

**Broader Impact Statement**

Our work introduces a family of foundation models particularly optimized for deployment on mobile and edge devices. As such, it facilitates broader use of foundation models and development of applications for wider user bases. MobileCLIP2 may be used for various applications such as image classification where its output is impacted by the existing biases of the training datasets and teacher models.

**Acknowledgments**

We would like to thank Albin Madappally Jose, Barry Theobald, Chen Huang, Rick Chang, and Apple Machine Learning Research team for their help and discussions throughout this project.

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

# A    Experimental Setup

Table 15 provides a summary of datasets used in our ablations and experiments.

Table 15: **Summary of pretraining datasets.**

| Dataset | Num. Samples | CLIP Teachers | Syn. Captioner | Num. Image Augs. | Num. Syn. Caps. | BFloat16 | Size (TBs) |
|---|---|---|---|---|---|---|---|
| DataComp-1B12M | 12.8M | ✗ | ✗ | ✗ | ✗ | ✗ | 0.9 |
| DFN-2B12M | 12.8M | ✗ | ✗ | ✗ | ✗ | ✗ | 0.2 |
| DFN-5B12M | 12.8M | ✗ | ✗ | ✗ | ✗ | ✗ | 1.5 |
| DataCompDR-12M | 12.8M | OpenAI + DataComp-XL | LAION-2B → MSCOCO-123k | 30 | 5 | ✓ | 2.1 |
| DFNDR-2B12M | 12.8M | DFN-2B + DFN-2B-s39B | DFN-2B → MSCOCO-38k | 30 | 5 | ✓ | 1.3 |
| DFNDR-5B12M | 12.8M | DFN-2B + DFN-2B-s39B | DFN-2B → MSCOCO-38k | 30 | 5 | ✓ | 2.6 |
| DataComp-1B | 1.3B | ✗ | ✗ | ✗ | ✗ | ✗ | 90 |
| DFN-2B | 1.9B | ✗ | ✗ | ✗ | ✗ | ✗ | 65 |
| DataCompDR-1B | 1.3B | OpenAI + DataComp-XL | LAION-2B → MSCOCO-123k | 10 | 5 | ✗ | 140 |
| DFNDR-2B | 1.9B | DFN-2B + DFN-2B-s39B | DFN-2B → MSCOCO-38k | 2 | 5 | ✗ | 162 |

Table 16 summarizes the hyperparameters we used to train MobileCLIP2. For training on 13B seen samples, we use either a setup with 32x8xA100-40GB GPUs or a setup with 16x8xH100-80GB GPUs. For our ablations we train for 30k seen samples using 4x8xH100-80GB GPUs and global batch size 8192.

Table 16: Training hyperparameters for our CLIP experiments on DFNDR-2B.

| Hyperparameter | S0 | S2 | B | S3 | S4 |
|---|---|---|---|---|---|
| Input resolution | $256^2$ | $256^2$ | $224^2$ | $256^2$ | $256^2$ |
| Context length | | | 77 | | |
| Data augmentation | | | RandAugment | | |
| Random resize crop scale | | | [0.08, 1.0] | | |
| Random resized crop ratio | | | [0.75, 1.33] | | |
| RangeAugment target value | | | (40, 20) | | |
| Train iterations | | | 200k | | |
| Warmup iterations | 10k | 10k | 2k | 2k | 2k |
| Global batch size | 65536 | 65536 | 65536 | 114688 | 114688 |
| Optimizer | | | AdamW | | |
| AdamW beta1 | | | 0.9 | | |
| AdamW beta2 | | | 0.95 | | |
| Max learning rate | | | 1e-3 | | |
| Min learning rate | 1e-6 | 1e-6 | 1e-6 | 0 | 0 |
| LR. decay schedule | | | cosine | | |
| Weight decay rate | | | 0.2 | | |
| Gradient clipping | | | 1.0 | | |
| Mixed precision | | | BFloat16 | | |
| EMA decay rate | 0.9995 | No EMA | No EMA | No EMA | No EMA |
| CLIP loss weight | 0.0 | 0.0 | 0.0 | 0.0 | 0.0 |
| KD loss weight | 1.0 | 1.0 | 1.0 | 1.0 | 1.0 |
| GT caption weight | | | 1.0 | | |
| Synth. caption weight | | | 1.0 | | |
| Synth. teacher | | | CoCa-ViT-L/14 - DFN-2B → MSCOCO-38k | | |
| Teacher 1 | | | DFN2B-CLIP-ViT-L-14-s39b | | |
| Teacher 2 | | | DFN2B-CLIP-ViT-L-14 | | |
| Teacher 1 logit scale | | | 70.0 | | |
| Teacher 2 logit scale | | | 60.0 | | |
| Teacher resolution | | | 224×224 | | |

## A.1    Training details for CoCa caption generators

We use OpenCLIP to train CoCa-ViT-L/14 architecture (`coca_ViT-L-14`). We pretrain models on DFN-2B and fine-tune on various datasets. Table 17 summarizes the hyperparameters for our CoCa pretraining and fine-tuning.

Table 17: Training hyperparameters for our CoCa models trained on DFN-2B.

| Hyperparameter | DFN-2B Pretrain | Fine-tune |
|---|---|---|
| Input resolution | $224^2$ | $224^2$ |
| Context length | 77 | 77, 255 |
| Seen samples | 12.8B | 12M |
| Train iterations | 195k | 3k, 6k |
| Early stop iterations | 143k | N/A |
| Warmup iterations | 10k | 1k |
| Global batch size | 65536 | 4092, 2048 |
| Optimizer | AdamW | |
| AdamW beta1 | 0.9 | |
| AdamW beta2 | 0.95 | |
| Max learning rate | 1e-3 | 1e-5 |
| Min learning rate | 0 | |
| LR. decay schedule | cosine | |
| Weight decay rate | 0.2 | 0.1 |
| Gradient clipping | 1.0 | |
| Mixed precision | amp | |
| CoCa caption loss weight | 2.0 | |
| CoCa contrastive loss weight | 1.0 | |
| GPU Setup | 32x8xA100-40GBs | 1x8xH100-80GBs |

## A.2 Training details for VLM

To assess the quality of the vision encoders, we adopt the LLaVA-1.5 (Liu et al., 2024a) training framework. This framework consists of two stages: (1) projector training, and (2) joint fine-tuning of the projector and the language model on 665K instruction-tuning samples. The hyperparameters used in our experiments are summarized in Table 18. For the language model, we use Qwen2-7B-Instruct Wang et al. (2024) as opposed to Vicuna-7B. In both the stages the vision encoder remains frozen.

| | Stage-1 | Stage-2 |
|---|---|---|
| Data | LLaVA-1.5 558K | LLaVA-1.5 665k |
| Learning Rate | 1e-3 | 2e-5 |
| Global Batch Size | 256 | 128 |
| Epochs | 1 | 1 |
| LR. schedule | cosine decay | cosine decay |
| LR. warmup ratio | 0.03 | 0.03 |
| Optimizer | AdamW | AdamW |
| Trainable modules | Projector | Projector + Language Model |

Table 18: LLaVA-1.5 training setup used in ablations for Table 9.

## A.3 Training details for dense prediction tasks

### A.3.1 Object detection

We train object detection models with MaskRCNN detection heads. Along with detection, these models also perform instance segmentation. We follow the settings prescribed in recent works like Liu et al. (2024c); Wei et al. (2023); Singh et al. (2023); Vasu et al. (2024b). All evaluations reported in the main paper are from single-scale evaluations on MS COCO validation set following prior works. We sweep through stochastic depth rate in steps of 0.05 and peak learning rate for all the results reported in the paper and the ranges are listed in Table 19. For ViT-B/16 models, we use ViTDet style feature pyramid network. For MCi architectures, we follow the setup described in Vasu et al. (2023b). All models were trained using MMDetection library Chen et al. (2019) on a single node with 8 A100 NVIDIA GPUs.

### A.3.2 Semantic Segmentation

We train segmentation models with UperNet and SemanticFPN heads. These models are trained on ADE20k Zhou et al. (2017) dataset following the settings prescribed in Liu et al. (2024c); Wei et al. (2023); Singh et al. (2023); Vasu et al. (2024b). All evaluations reported in the main paper are from single-scale evaluations on validation set following prior works. For ViT-B/16 models, we use ViTDet style feature pyramid network with UperNet head. For MCi architectures, we follow the setup described in Vasu et al. (2023b) and train models with only SemanticFPN head. We sweep through stochastic depth rate in steps of `0.05` and peak learning rate for all the results reported in the paper and the ranges are listed in Table 20. All models were trained using MMSegmentation library Contributors (2020) on a single node with 8 A100 NVIDIA GPUs.

### A.3.3 Depth Estimation

We follow the experimental setup and architecture as described in Wei et al. (2023); Vasu et al. (2024b). The models are trained and evaluated on NYUv2 dataset Nathan Silberman & Fergus (2012). We sweep through stochastic depth rate in steps of `0.05` and peak learning rate for all the results reported in the paper and the ranges are listed in Table 21.

Table 19: Training hyperparameters for object detection and instance segmentation experiments on MS COCO. "RRC" is `RandomResizedCrop`. We sweep through stochastic depth rate in steps of `0.05`.

| Hyperparameters | MaskRCNN |
|---|---|
| Stochastic depth rate | [0.0, ..., 0.3] |
| Data augmentation | Multi scale RRC |
| Train epochs | 12 |
| Batch size | 16 |
| Optimizer | AdamW |
| Peak learning rate | [5e-4, 2e-4, 1e-4] |
| LR. decay schedule type | Step-wise |
| LR. decay schedule | [8, 11] |
| Weight decay rate | 0.1 |

Table 20: Training hyperparameters for semantic segmentation experiments on ADE20k. "RRC" is `RandomResizedCrop`. We sweep through stochastic depth rate in steps of `0.05`.

| Hyperparameters | UperNet | SemanticFPN |
|---|---|---|
| Stochastic depth rate | | [0.0, ..., 0.2] |
| Data augmentation | | RRC |
| Crop Size | | 512×512 |
| Train iterations | 160k | 40k |
| Batch size | 16 | 64 |
| Optimizer | | AdamW |
| Peak learning rate | | [5e-4, 2e-4, 1e-4] |
| LR. decay schedule type | | Polynomial |
| Warmup iterations | 1500 | - |
| Weight decay rate | 0.01 | 5e-4 |

Table 21: Training hyperparameters for depth estimation experiments on NYUv2 dataset. "RRC" is `RandomResizedCrop`. We sweep through stochastic depth rate in steps of `0.05`.

| Hyperparameters | Value |
|---|---|
| Stochastic depth rate | [0.0, ..., 0.2] |
| Data augmentation | RRC |
| Crop Size | 480×480 |
| Train epochs | 25 |
| Batch size | 24 |
| Optimizer | AdamW |
| Peak learning rate | [7e-4, 5e-4, 2e-4, 1e-4] |
| Layer decay rate | 0.8 |
| Weight decay rate | 0.05 |

