# OpenReview forum: "MobileCLIP2: Improving Multi-Modal Reinforced Training"
_TMLR — Accepted by TMLR_

### Review · Reviewer_4B8F · 2025-05-13

**Summary Of Contributions:**

This paper introduces a new family of low latency image-text foundation models, MobileCLIP2. Building upon MobileCLIP, MobileCLIP2 improves multi-modal reinforced training by training on a better pretraining dataset (DFN-5B) and utilizing stronger CLIP teachers as well as an improved synthetic caption generator (CoCa). It also introduces two new architectures, L and XL, by adding an additional transformer stage. The resulting MobileCLIP2 family achieves state-of-the-art performance at a range of latencies, especially on zero-shot image classification tasks including ImageNet-1K and 38 datasets from DataComp.

**Audience:**

Yes

**Broader Impact Concerns:**

The paper has addressed broader impact concerns.

**Claims And Evidence:**

Yes

**Requested Changes:**

Please fix typos (e.g., "multi-resolution training (Li et al., 2023a) proposed training at for cost-effective training" in Section 5) and answer the following questions.

- Section 5 "Improving the training of multi-modal models focus on three aspects: data, objective function and architecture. Our MobileCLIP2 builds on MobileCLIP and provides improvements in all three aspects.": What are improvements in the objective function compared to MobileCLIP?
- Table 4 caption "CLIP teachers are the same as MobileCLIP (OpenAI+DataComp-XL CLIPViT-L/14): This should be removed (it seems to have been taken directly from Table 5 caption).

**Strengths And Weaknesses:**

Strengths
- The paper demonstrates the effectiveness of improved training components, such as pretraining dataset, distillation framework and new architectures, over MobileCLIP through a systematic ablation study.
- MobileCLIP2 achieves superior performance on zero-shot image classification tasks compared to previous approaches while maintaining low latency and a small model size.
- The detailed description of hyperparameters in the Appendix ensures the reproducibility of this work.

Weaknesses

There are no major weaknesses worth pointing out. The lower performance on retrieval tasks also seems attributable to biases in the training dataset (DFN-5B), as the authors noted in the main draft. One suggestion for further experimentation is to evaluate a key motivation behind joint training of image and text representations, that is, such vision encoders can be paired with pretrained LLMs to build performant and versatile vision-language models. Given the growing interest in multimodal LLMs that can run on mobile and edge devices, it would have been more compelling if the authors had demonstrated how the MobileCLIP2 vision encoder performs when integrated with 1B or 3B scale LLMs across various VQA tasks.

---

> ### Author Response · Authors · 2025-05-29
> **Thank you for your review**
>
> We thank the reviewer 4B8F for their feedback and noting strengths such as effectiveness of MobileCLIP2 model family and the detailed descriptions. We hope to address their concerns below and will update the manuscript with their feedback in our revision.
>
> # Evaluation in LLaVa1.5 setup
> Thank you for noting the importance of including additional VLM evaluations. Below we report vision-language evaluations using MobileCLIP-2 pretrained models in the LLaVA 1.5 setup [1]. We keep the vision backbone frozen for all the runs and use Qwen-7B instead of Vicuna-7B. All other training details are the same as the original LLaVA 1.5 setup. We evaluate ViT-B/16 models pretrained on DataComp (DC), DFN, DataCompDR (DCDR), and DFNDR for 13B seen samples. We observe that on average training on DFNDR achieves 3.5% higher accuracy compared with DFN pretrained model, 1.6% better than DataComp pretrained model, and 0.6% better than DataCompDR pretrained model. These results demonstrate that training on DFNDR is not just more efficient but also reaches better results in a diverse set of downstream evaluations.
>
> |       | Avg  | GQA  | SQA  | TextVQA | POPE | MMMU | MMB  | VizWiz | VQAv2 |
> |:-----:|:----:|:----:|:----:|:-------:|:----:|:----:|:----:|:------:|:-----:|
> | DC    | 61.0 | 59.6 | 71.5 | **50.5**    | 81.8 | 42.6 | 59.1 | 51.8   | 70.7  |
> | DFN   | 59.1 | 56.9 | 71.3 | 46.0    | 81.4 | 41.9 | 52.2 | **56.1**   | 66.9  |
> | DCDR  | 62.0 | 60.3 | **73.1** | 50.4    | 81.7 | 43.6 | 60.2 | 54.9   | 72.1  |
> | DFNDR | **62.6** | **60.4** | 72.9 | 49.9    | **83.3** | **45.2** | **61.9** | 54.5   | **72.4**  |
>
> [1] Liu, Haotian, et al. "Improved baselines with visual instruction tuning." Proceedings of the IEEE/CVF Conference on Computer Vision and Pattern Recognition. 2024.
>
> # Objective Function
> > What are improvements in the objective function compared to MobileCLIP?
>
> We apologize for not making this clear. Our objective function is the same as MobileCLIP but we uncover the importance of additional tuning of the logit scaling of teacher models.
> More specifically, as discussed in Section 2.3 and Table 3, we observe that the value of $\tau_k$ (logit-scale) in Eq. 1 needs to be tuned for each model. MobileCLIP used the values stored in the model checkpoints (100 for both `openai` and `datacomp_xl_s13b_b90k` teachers) while we find values as small as 50 can be optimal which translates to smoothening the similarity matrix of the teacher. These findings are complementary to our improvements to the dataset through better teacher models, synthetic caption generators, and architecture.
>
> # Typos
> Thank you for reporting the typos. We have fixed them and reviewed the paper for any remaining typos that will appear in a revision.
>
> > Table 4 caption "CLIP teachers are the same as MobileCLIP (OpenAI+DataComp-XL CLIPViT-L/14): This should be removed (it seems to have been taken directly from Table 5 caption).
>
> Thank you for catching this error. We have fixed it and will appear in our revision.

---

### Review · Reviewer_cmuV · 2025-05-19

**Summary Of Contributions:**

This paper presents MobileCLIP2, a new family of efficient and low-latency image-text models. It improves upon MobileCLIP through enhanced multi-modal reinforced training. Specifically, the authors investigate the effectiveness of DFN-pretrained models as teachers compared to those pretrained on DataComp. Then, they ablate serval critical aspects of the caption generator, including the pretraining and fine-tuning datasets, the number of synthetic captions, and the context length of CoCa models. Based on these analyses, the authors select the best solution and build MobileCLIP2, which achieves state-of-the-art zero-shot performance on ImageNet-1k at low latency.

**Audience:**

Yes

**Broader Impact Concerns:**

None.

**Claims And Evidence:**

Yes

**Requested Changes:**

See weakness. It is crucial to demonstrate that the newly proposed visual encoder not only achieves superior performance on standard vision benchmarks but also provides broader benefits, such as enhancing multimodal large language models. Therefore, I recommend including such experiments to further strengthen the paper.

**Strengths And Weaknesses:**

Strengths:
1. The paper propose a new family of efficient and low-latency image-text models through improved multi-modal reinforced training.
2. The authors find that without fine-tuning, caption generators lead to worse retrieval performance, while additional fine-tuning on datasets like MSCOCO significantly improves results.
3. Extensive experiments demonstrate the model's efficency.
4. The authors commit to releasing their data generation code and pretrained models to support scalable and reproducible research.


Weakness:
1. The paper lacks quantitative discussion of training efficiency compared to baseline methods.
2. Given that modern vision encoders are frequently integrated into large-scale vision-language models (e.g., CLIP, SigLIP), it would be beneficial to include experiments demonstrating the advantages of replacing the vision encoder in models such as LLaVA with MobileCLIP-2, highlighting any improvements over baseline performance.

---

> ### Author Response · Authors · 2025-05-29
> **Thank you for your review**
>
> We thank the reviewer cmuV for their feedback and noting strengths such as effectiveness of MobileCLIP2 model family and the extensive experiments. We hope to address their concerns below and will update the manuscript with their feedback in our revision.
>
> # Training Efficiency
>
> Thank you for noting the importance of discussing the training efficiency. Here we report the efficiency gain compared to various baselines and will include plots in a revision of the paper that compare the performance during training. Training on DFNDR-12M for 30M seen samples is **5x more efficient** than training on DataComp-12M, i.e., we reach the ImageNet-1k zero-shot accuracy achieved by training for 30M seen samples on DataComp-12M after seeing only 6M seen samples of DFNDR-12M. Similarly, training on DFNDR-12M is 3.33x more efficient compared to DFN2B-12M and 1.33x more efficient compared with DataCompDR-12M. We also observe 1.66x speedup when training on DFNDR-2B compared with training on DataCompDR-1B for 13B seen samples.
>
> Please note that similar to DataCompDR, training on DFNDR datasets do not have any wall-clock time overhead, i.e., each training step of training on DataComp, DFN, DataCompDR, and DFNDR takes the same amount of time. That means any efficiency gains in terms of the number of samples and training iterations directly translate to wall-clock time efficiency gains.
>
> # Evaluation in LLaVa1.5 setup
>
> Thank you for noting the importance of including additional VLM evaluations. Below we report vision-language evaluations using MobileCLIP-2 pretrained models in the LLaVA 1.5 setup [1]. We keep the vision backbone frozen for all the runs and use Qwen-7B instead of Vicuna-7B. All other training details are the same as the original LLaVA 1.5 setup. We evaluate ViT-B/16 models pretrained on DataComp (DC), DFN, DataCompDR (DCDR), and DFNDR for 13B seen samples. We observe that on average training on DFNDR achieves **3.5% higher accuracy** compared with DFN pretrained model, 1.6% better than DataComp pretrained model, and 0.6% better than DataCompDR pretrained model. These results demonstrate that training on DFNDR is not just more efficient but also reaches better results in a diverse set of downstream evaluations.
>
> |       | Avg  | GQA  | SQA  | TextVQA | POPE | MMMU | MMB  | VizWiz | VQAv2 |
> |:-----:|:----:|:----:|:----:|:-------:|:----:|:----:|:----:|:------:|:-----:|
> | DC    | 61.0 | 59.6 | 71.5 | **50.5**    | 81.8 | 42.6 | 59.1 | 51.8   | 70.7  |
> | DFN   | 59.1 | 56.9 | 71.3 | 46.0    | 81.4 | 41.9 | 52.2 | **56.1**   | 66.9  |
> | DCDR  | 62.0 | 60.3 | **73.1** | 50.4    | 81.7 | 43.6 | 60.2 | 54.9   | 72.1  |
> | DFNDR | **62.6** | **60.4** | 72.9 | 49.9    | **83.3** | **45.2** | **61.9** | 54.5   | **72.4**  |
>
> [1] Liu, Haotian, et al. "Improved baselines with visual instruction tuning." Proceedings of the IEEE/CVF Conference on Computer Vision and Pattern Recognition. 2024.

---

### Review · Reviewer_hu5S · 2025-06-01

**Summary Of Contributions:**

Proposes MobileCLIP2, a family of CLIP models built on top of MobileCLIP, achieving state-of-the-art zero-shot accuracy on ImageNet-1k. Key improvements include leveraging stronger CLIP teachers, training better captioning models, and conducting a comprehensive study on tuning and ensembling methods.

**Audience:**

Yes

**Claims And Evidence:**

Yes

**Requested Changes:**

There are some formatted issues in a few table headers (like table 8)

**Strengths And Weaknesses:**

Strengths: The experiments are comprehensive and the descriptions are clear. The paper is easy to follow.

---

> ### Author Response · Authors · 2025-06-11
> **Thank you for your review**
>
> We thank reviewer hu5S for their feedback and recognizing the strength of our comprehensive experiments and description of the paper. We have fixed the formatting issues in the new revision.

---

### Author Response · Authors · 2025-06-11
**We thank the reviewers for their thoughtful feedback and upload a revision**

We thank reviewers for their thoughtful feedback. We hope our responses have addressed their concerns. We have uploaded a new revision of the paper with the following changes:
- New Figure 2 along with description in the text demonstrates up to 5x training efficiency using our proposed datasets and training recipe.
- New Section 4.2 and Table 14: VLM evaluations in LLaVa-1.5 setup that demonstrate up to 3.5% higher accuracy.
- Minor formatting corrections to Tables.

---

### Decision · Action_Editor_dfK7 · 2025-07-20

**Recommendation:** Accept as is

**Audience:**

Yes

**Audience Explanation:**

MobileCLIP2 is a new family of image-text models that achieve state-of-the-art performance (e.g., ImageNet zero-shot) while maintaining small model sizes and low latency, making them well-suited for mobile and edge devices. Given the growing interest in multi-modal research, MobileCLIP2 is likely to be of broad interest to the community.

**Claims And Evidence:**

Yes

**Claims Explanation:**

In this work, the authors propose MobileCLIP2, a family of mobile/edge device-friendly image-text models. MobileCLIP2 improves the multi-modal reinforced training of MobileCLIP by utilizing the better trained CLIP and Captioner (CoCa) teachers on DFN dataset. The resulting MobileCLIP2 achieves state-of-the-art performance with various compute budgets.

Initially, reviewers were concerned about the lack of VLM experiments and clarifications on method details (e.g., training efficiency and objective). The provided rebuttal effectively assuaged the reviewer concerns. In the end, all three reviewers recommend accepting the paper. Additionally, the provided experiments effectively verify the design choices and support the claims made in the submission. After carefully checking the submission, reviews, and author rebuttal, the Action Editor thus agrees with the reviewers' recommendation.